# Myocardial Fibrosis in Hypertrophic Cardiomyopathy: A Perspective from Fibroblasts

**DOI:** 10.3390/ijms241914845

**Published:** 2023-10-02

**Authors:** Maja Schlittler, Peter P. Pramstaller, Alessandra Rossini, Marzia De Bortoli

**Affiliations:** Eurac Research, Institute for Biomedicine (Affiliated to the University of Lübeck), 39100 Bolzano, Italy

**Keywords:** hypertrophic cardiomyopathy, myocardial fibrosis, cardiac fibroblast, myofibroblasts, TGF-β

## Abstract

Hypertrophic cardiomyopathy (HCM) is the most common inherited heart disease and the leading cause of sudden cardiac death in young people. Mutations in genes that encode structural proteins of the cardiac sarcomere are the more frequent genetic cause of HCM. The disease is characterized by cardiomyocyte hypertrophy and myocardial fibrosis, which is defined as the excessive deposition of extracellular matrix proteins, mainly collagen I and III, in the myocardium. The development of fibrotic tissue in the heart adversely affects cardiac function. In this review, we discuss the latest evidence on how cardiac fibrosis is promoted, the role of cardiac fibroblasts, their interaction with cardiomyocytes, and their activation via the TGF-β pathway, the primary intracellular signalling pathway regulating extracellular matrix turnover. Finally, we summarize new findings on profibrotic genes as well as genetic and non-genetic factors involved in the pathophysiology of HCM.

## 1. Introduction

Hypertrophic cardiomyopathy (HCM) is a genetic heart disorder and the leading cause of sudden cardiac death in young people [1]. HCM affects approximately 1 out of 500 people in the general population and is inherited as an autosomal dominant trait with incomplete, age-related penetrance and variable disease expressivity [2,3]. Around 60% of all patients have a family history of HCM, and more than half of them harbour a variant in one of the genes encoding structural components of the cardiac sarcomere [4,5,6]. The two most frequently mutated genes are *MYH7* and *MYBPC3*, which encode β-myosin heavy chain and myosin binding protein C, respectively [7]. Approximately 5–7% of all HCM mutation carriers exhibit double heterozygous, compound heterozygous, or homozygous mutations, which are typically associated with an earlier onset and a more severe form of HCM [6,8,9]. In recent years, this percentage has increased to 8–9% thanks to the next-generation sequencing, which is now established as the gold standard for the diagnosis of hereditary disorders [9]. Causative variants for HCM were not only found in structural proteins of the sarcomere but also proteins regulating sarcomere function and calcium homeostasis [8,9]. At least 31 genes have been associated with HCM (Table 1). Additional genes are involved in syndromic HCM, such as Anderson–Fabry disease and glycogen storage diseases, and specific genetic screening, histological examinations, and metabolic testing are necessary to make a differential diagnosis [10,11]. The majority of the known genetic causes of HCM are dominant-negative missense mutations [8,9]. Haplo-insufficiency conditions caused by splice sites, non-sense variants, and insertions/deletions leading to a frameshift and premature stop codon creation are more frequently described for the *MYBPC3* gene [8,9]. Despite the identification of these mutations, more than half of all HCM patients remain without a clear genetic diagnosis, potentially carrying mutations in unidentified genes or having non-genetic risk factors [12].

HCM has a complex pathophysiology and a heterogeneous clinical profile [1]. The most characteristic feature of the disease is left ventricular hypertrophy, which typically develops during adolescence, although cases of onset in the sixth or seventh decade of life have been reported as well [14,15]. The distribution of the hypertrophic tissue is often asymmetric and can vary considerably even between closely related patients [14,15]. At the cellular level, HCM is characterized by cardiomyocyte hypertrophy and sarcomeric disarray [14]. In addition, the hearts of patients with HCM display areas of myocardial fibrosis, which manifests as an excess accumulation of extracellular matrix (ECM) proteins. The ECM of mammalian hearts consists of fibrillar proteins, such as collagen type I and III, that provide structural support to the heart and facilitate force transmission and non-fibrillar proteins, including glycoproteins, glycosaminoglycans, and proteoglycans that provide storage for latent growth factors and proteases that are released when required [16,17]. In animal models of HCM, fibrillar proteins are consistently upregulated [18,19,20,21] and there is also convincing evidence from human studies for an increased collagen turnover in HCM. Shirani et al., for instance, investigated the hearts of young HCM patients who died of sudden cardiac death and found an eight-fold increase in cardiac collagen compared to controls [22]. Other studies measured serological markers of collagen turnover in patients with HCM and also found significantly elevated levels compared to controls [23,24]. Although less studied than the fibrillar proteins, there is also growing evidence for an upregulation of non-fibrillar ECM proteins in HCM [16]. 

Myocardial fibrosis is a common feature of many types of cardiac disease. However, it is important to note that genetic cardiomyopathies show distinct types and patterns of myocardial fibrosis [25]. Hearts from patients with arrhythmogenic cardiomyopathy, for example, are characterized by a loss of ventricular myocardium with subsequent fibro-fatty tissue replacement [26]. They also display myocardial fibrosis, which is mainly present in the right ventricle (RV) outflow tract, RV free wall, and RV wall of the septum [25]. In HCM, on the other hand, fibrosis is typically located in the septum, and/or the left ventricle and RV insertion points [25].

Myocardial fibrosis is generally considered a secondary response to cardiomyocyte death or myocardial remodelling [27]. However, in the case of HCM, there is growing evidence for the activation of fibrotic pathways already early during the course of the disease before hypertrophic remodelling occurs. Kim et al., for instance, observed an upregulation of profibrotic genes in young mice carrying an HCM mutation in the α-MHC gene before they developed hypertrophy [28]. These results were also confirmed in humans by Ho et al., who found biomarkers of increased collagen synthesis in HCM mutation carriers before any left ventricular hypertrophy was evident [24]. These findings suggest that in HCM, myocardial fibrosis is not merely a secondary reaction to hypertrophic remodelling but potentially an important contributor to the aetiology of the disease. Given the difference between reparative fibrosis observed in other cardiac conditions and the presence of fibrosis at specific sites of cardiac tissue in HCM, it seems important to consider the evidence on myocardial fibrosis separately for HCM. 

While typical HCM mutations in sarcomere proteins have been shown to affect cardiomyocyte hypertrophy, calcium signalling, autophagy, and glucose oxidation [12], less is known about how these mutations can lead to cardiac fibrosis. The cardiac fibroblasts and their activation via transforming growth factor-β (TGF-β) signalling were reported as potential mediators of fibrosis during HCM pathogenesis [12], and although cardiomyocytes are usually the main cell type structurally and functionally affected in genetic cardiomyopathies, a full investigation of cardiac fibroblasts, their function, and activation could improve the understanding of the molecular mechanisms leading to fibrosis in HCM.

In this review, we therefore summarize and discuss the latest evidence on the cellular and genetic effectors involved in the development of myocardial fibrosis, specifically in HCM, as well as the TGF-β pathway, the most important molecular signalling pathway. 

## 2. Cellular Effectors of Myocardial Fibrosis

Depending on the region, 15–24% of the cells of a healthy adult heart are cardiac fibroblasts (cFB) [29]. Under physiological conditions, cFB provides structural support to the heart by regulating the synthesis and degradation of ECM proteins and enzymes. Upon physiological or pathophysiological stimuli, cFB proliferate, migrate, and differentiate into myofibroblasts (MyoFB), cells that express contractile proteins, such as alpha-smooth muscle actin (α-SMA), and secrete large amounts of ECM proteins and enzymes involved in ECM remodelling, including different types of collagen, matrix metalloproteinases, and tissue inhibitors of metalloproteinases [27]. In conditions that involve cardiomyocyte death, such as myocardial infarction, cFB migrates to the location of the injury, differentiates into MyoFB, and forms scar tissue to replace the necrotic areas, a process that is referred to as replacement fibrosis. As the regenerative capacity of the myocardium is limited, cFB activation is a crucial mechanism to maintain the structural integrity of the heart in case of injury [17]. However, in HCM, sustained activation of MyoFB occurs in the absence of cardiomyocyte death and leads to an excessive accumulation of ECM proteins in the interstitial space, a condition known as reactive or interstitial fibrosis [27]. Interstitial fibrosis may severely affect cardiac function on several levels: fibrotic tissue increases stiffness and reduces the compliance of the heart, which leads to diastolic dysfunction [16,23], and the accumulation of ECM proteins around cardiomyocytes may disrupt the conduction of the electrical impulse and lead to re-entry circuits and dysrhythmias [30].

The activation of MyoFB can be triggered by several different stimuli, including growth factors, cytokines, mechanical stress, and neurohumoral pathways, such as renin-angiotensin-aldosterone signalling or β-adrenergic stimulation [27]. Activation of renin-angiotensin-aldosterone signalling is a common finding in hearts with myocardial fibrosis, independent of the underlying cause [27]. Angiotensin II binds to the angiotensin type 1 receptor on the surface of the cFB and activates downstream signalling pathways that directly or indirectly, via TGF-β, induce the trans-differentiation into MyoFB [31]. 

Immune cells, endothelial cells, cardiomyocytes, and cFB themselves may all secrete growth factors and cytokines, such as TGF-β, TNF-α, IL-1, IL-10, and PDGFs, that bind to surface receptors on the cFB and initiate intracellular signalling cascades that induce the expression of α-SMA and regulate the expression of genes involved in ECM turnover [27]. The multifunctional cytokines of the TGF-β family are by far the most extensive activators of cFB and the role of their downstream signalling pathways for the development of myocardial fibrosis in HCM will be discussed in detail in the following sections. 

At last, cFB may also trans-differentiate in response to changes in the stiffness of the ECM via mechanosensitive receptors that activate profibrotic cascades [27]. One of these pathways involves the activation of focal adhesion kinases and signalling via integrins. While blocking focal adhesion kinases reduces the development of myocardial fibrosis, it is not entirely clear whether this is attributed to the activation of cFB or the effects of focal adhesion kinase activation on other cells [32]. 

## 3. TGF-β Signalling

### 3.1. TGF-β Signalling in HCM Myocardial Fibrosis

TGF-β signalling pathways play a central role in the development of fibrosis in various tissues [17]. Mammals express three isoforms of TGF-β, which are encoded by three distinct genes [33]. In the heart, several types of cells, including cardiomyocytes, cFB, macrophages, and vascular cells, can produce TGF-β depending on the pathologic stimulus [17,34]. TGF-β is secreted in its latent form, which is unable to interact with its receptors and can therefore be stored in the tissue [33]. The activation of latent TGF-β is an important checkpoint for TGF-β signalling and to date, the molecular mechanisms for its activation are not well understood, particularly in the cardiac context. In other tissues, it has been described that the latent TGF-β is activated through proteolytic cleavage by a wide range of proteases (specifically named sheddases), including thrombospondin 1, matrix metalloproteinases, and activated integrins [35,36]. A few papers have described Furin, ADAMTS16, ADAMTS4, and LTBPs as sheddases able to activate latent TGF-β in cardiac fibroblasts [37,38,39,40]. After proteolytic cleavage, bioactive TGF-β is released, which then binds to heterodimeric TGF-β receptors (TβR-I/TβR-II) on the surface of target cells. The Association of TGF-β with the TβR complex initiates profibrotic cascades via canonical or non-canonical signalling pathways [41]. The canonical TGF-β pathway involves the phosphorylation of Smad2 and 3, which, after association with Smad4, translocate to the nucleus and regulate the transcription of profibrotic genes. Smad-independent, non-canonical TGF-β signalling comprises several different pathways, including mitogen-activated protein kinase (MAPK), phosphoinositide-3-kinase, and Rho GTPases (Rho) pathways, which eventually also alter the expression of fibrotic genes [34,42]. In the context of HCM, evidence from human and animal studies suggests an association between the upregulation of the different TGF-β isoforms and myocardial fibrosis. In a recent study, for instance, plasma proteomic profiling and subsequent pathway analyses revealed an induction of the TGF-β pathway in HCM patients [43], and another study found significantly increased serum TGF-β1 levels in patients with HCM compared to controls [38]. Among patients with HCM, higher TGF-β1 levels were also correlated with worse clinical outcomes [44]. Li et al. showed that the upregulation of the TGF-β1 gene and protein expression is a regional phenomenon that occurs specifically in hypertrophied areas of HCM hearts [45]. These findings were confirmed by immunohistochemistry analyses of myocardial tissue of patients with HCM, where TGF-β1 expression was significantly increased in areas of interstitial fibrosis [46]. The induction of TGF-β pathways in HCM has also been confirmed by Liu et al., who performed a pathway analysis of publicly available microarray data from patients and controls. Surprisingly and in contrast to other papers, this study found an upregulation of TGFB2 and TGFB3 expression in HCM, whereas TGFB1 was not altered [18], which might be related to different temporal expression of the three isoforms [17]. The upregulation of TGF-β in HCM-associated myocardial fibrosis has also been confirmed in numerous studies using animal models. Mice with hypertrophy and myocardial fibrosis induced by HCM mutations in the α-MHC gene, for example, displayed upregulated TGF-β gene expression specifically in the non-cardiomyocyte cell population [19]. The administration of TGF-β-neutralizing antibodies in these mice attenuated the development of fibrosis, suggesting a central role of TGF-β signalling in the development of fibrosis [19]. In a pressure-overload model in rats, cFB activation and myocardial fibrosis coincided with increased TGF-β gene expression, and these effects were also prevented by TGF-β neutralizing antibodies [47]. Transgenic mice overexpressing TGF-β1 developed cardiac hypertrophy and interstitial fibrosis [48], whereas the deletion of one TGF-β1 allele attenuated age-induced cardiac fibrosis in heterozygous TGF-β1 (+/−) mice [49]. Lucas et al. completely interrupted TGF-β signalling in mice by overexpressing an inducible dominant-negative mutation of the TβR-II. While these mice were protected from MyoFB proliferation and fibrosis, they also developed left ventricular dilation and dysfunction in response to pressure overload, which underlines the fine balance between ECM secretion and degradation for proper cardiac function [50]. In summary, there is convincing evidence for the local and systemic upregulation of TGF-β in patients with HCM, and mechanistic animal studies have also shown a clear association between the induction of TGF-β and myocardial fibrosis. Although the different TGF-β isoforms have similar activating effects on cFB in vitro, their relative roles in the development of myocardial fibrosis in vivo are still controversial [17]. Of note, while the above-mentioned studies indicate an increased gene or protein expression of TGF-β, direct evidence on the amount of bioactive TGF-β in HCM is missing [34]. However, Khan et al. demonstrated that levels of bioactive TGF-β correlate better with functional outcomes of fibrosis than total TGF-β in patients with heart failure due to ischemic and dilated cardiomyopathy, and therefore, a direct assessment of TGF-β activity in patients with HCM would be highly relevant [30].

### 3.2. Canonical and Non-Canonical TGF-β Signalling

Several in vivo and in vitro studies have investigated the relative contribution of canonical and non-canonical TGF-β pathways to the development of fibrosis. Khalil et al. used an inducible transgenic mouse model to inhibit canonical signalling by deleting Smad2 and/or Smad3 in selected cell types. The deletion of Smad2/3 and Smad3 in the cFB population significantly reduced pressure overload- and TGF-β-induced fibrosis, whereas the deletion of Smad2/3 from cardiomyocytes had no effect on the development of fibrosis [51]. These results suggest a central role of cFB-specific canonical TGF-β signalling in the fibrogenic cascade. The involvement of Smad-dependent pathways was confirmed by Engebretsen et al., who treated mice with a small molecular inhibitor of TβR-I, which significantly reduced the pressure overload-induced phosphorylation of Smad2 and the development of fibrosis. However, while the treated mice had improved diastolic function and cardiac output, they also displayed left ventricular dilation and lesions of the cardiac valves [52]. Similar observations have been made by Russo et al., who created a mouse model carrying a MyoFB-specific knockout of Smad3. Acute pressure overload in these mice caused ECM degradation and cardiac dysfunction, demonstrating the importance of ECM homeostasis for proper heart function as well as the crucial role of canonical TGF-β pathways for the preservation of the ECM [53]. 

Of note, Meng et al. created a chronic HCM mouse model by genetically inducing the expression of a mutated cMyBP-C fragment in cardiomyocytes, which causes cardiac hypertrophy, myocardial fibrosis, and heart failure over a longer period of time than acute pressure-overload models and thus more closely mimics human HCM. During the early stages of the disease, both canonical and non-canonical TGF-β pathways were induced, whereas in chronic fibrosis, only Smad-dependent TGF-β signalling was upregulated. Ablation of TβR-II alleles specifically in MyoFB also blocked the phosphorylation of Smad3, attenuated myocardial fibrosis, and preserved cardiac function in the early stages of HCM. In advanced disease stages, TβR-II ablation even reverses existing fibrosis, making it an attractive candidate to pharmaceutically target myocardial fibrosis [54]. 

Although less studied than the canonical pathway, non-canonical TGF-β signalling has also been implicated in the development of myocardial fibrosis in HCM. Razzaque et al., for instance, generated transgenic mice overexpressing an inducible cardiomyocyte *MYBPC3* mutation. These mice displayed cardiomyocyte hypertrophy, disarray, and fibrosis as well as activation of non-canonical MEK-ERK signalling pathways. Injection of a specific inhibitor of MAPK/ERK reduced hypertrophy and improved cardiac function and survival of the mice, suggesting involvement of MEK-ERK signalling in the development of the pathology [55]. Using the same mouse model, Meng et al. demonstrated that inhibition of the p38/MAPK-activated protein kinase 2 pathway inhibited α-SMA expression in cFB, reduced the deposition of ECM, and alleviated myocardial fibrosis. In addition, the treatment reduced cardiac hypertrophy and prolonged survival of the mice [56]. The induction of non-canonical TGF-β signalling during the development of fibrosis in HCM was confirmed by a subsequent study that revealed a significant upregulation of TAK1 and its downstream target, p38, in HCM mice [54]. Intriguingly, the induction of non-canonical TGF-β signalling was only observed in the early stages of myocardial fibrosis, suggesting a less important role once fibrosis is established [54].

Taken together, these studies have presented convincing evidence for the critical role of TGF-β signalling in the fibrotic cascade in HCM. During the onset of fibrosis, both canonical and non-canonical signalling are involved, whereas in later, chronic stages of fibrosis, the canonical TGF-β pathway seems to be dominant. Figure 1 graphically summarizes the canonical and non-canonical TGF-β signalling pathways. 

## 4. Interactions between Cardiomyocytes and Cardiac Fibroblasts in HCM Myocardial Fibrosis

HCM is primarily considered a disease of the cardiomyocytes, and most of the mutations that have been associated with the disease affect sarcomere proteins that are highly expressed in cardiomyocytes. Traditionally, fibrosis in HCM has been considered a secondary response to cardiomyocyte hypertrophy; however, there is growing evidence for the early induction of profibrotic signalling in HCM [24,26]. However, the connection between the sarcomere mutations and the activation of profibrotic signalling in cFB remains well established. 

Some studies have been able to show a possible link between sarcomere protein mutations and myocardial fibrosis. Of note, several authors have observed greater fibrosis in HCM patients with sarcomere mutations than in those negative for genetic tests [57,58,59,60,61]. Moreover, positive mutation carriers without left ventricular hypertrophy were also shown to have fibrosis [60,62], with elevated serum levels of procollagen I C-terminal propeptide [22], suggesting that a profibrotic state could be induced by mutations and the fibrotic remodelling may be an early disease manifestation [22,57]. An association between HCM mutations and fibrosis was also supported by Teekakirikul et al., who showed that the expression of sarcomere protein mutations in myocytes activates proliferative and profibrotic signals in non-myocyte cells that induce pathologic remodelling in HCM [19]. It has also been speculated that altered biomechanical forces resulting from sarcomere protein mutations could provide a local mechanism for activating resident non-myocyte cells [19]. Finally, a recent study has also demonstrated that contrary to general belief, primary cFB actually does express myosin-binding protein C and other myofilament proteins [63]. Notably, a *MYBPC3* deficiency in these cells can promote their trans-differentiation into MyoFB via the NF-κB/TGF-β1/HIF-1α/aerobic glycolysis signal cascade, contributing to fibrosis in HCM [56].

## 5. Profibrotic Genes Cascade in HCM

The most prominent genes involved in ECM turnover and subsequent cardiac fibrosis encode for α-SMA, collagens, periostin, integrins, matrix metalloproteinases, and their tissue inhibitors [19,54,64]. In addition, some miRNAs and lncRNAs were reported to be associated with myocardial fibrosis in HCM [65,66,67]. Recent advances in single-cell RNA sequencing have allowed us to get a clearer picture of the interactions between different cell populations in a specific tissue. By using this technique, a deep alteration of intercellular communication, in particular between cardiomyocytes and cFB, through changes in ECM components was described in HCM [20]. Of note, Larson et al. showed altered ligand-receptor pair gene expression associated with ECM in cardiomyocytes and cFB from HCM patients [20]. Briefly, a substantial reduction in ligand expression, specifically interacting with the ECM, was described in HCM cardiomyocytes and cFB, suggesting that the dysfunctional myocytes lead to abnormal interaction with the ECM, which in turn propagates to cFB through changes in ECM component expression and interaction [20]. 

An overview of genes showing altered expression in HCM cardiac fibrosis is provided in Table 2. Several differentially expressed genes were listed. Differences among studies are probably due to the different analysis methods and HCM models used. 

## 6. Novel HCM Genetic and Nongenetic Actors 

The presence of a sarcomere mutation and a young age at diagnosis was reported as powerful predictors of adverse outcomes by the Sarcomeric Human Cardiomyopathy Registry (SHaRe) study, curated by 8 international HCM specialized centres [69]. Myocardial fibrosis is often related to an adverse outcome [69]; however, as mentioned above, there is a lack of knowledge on how a sarcomere pathogenic variant can lead to the fibrotic phenotype. Furthermore, we have to consider that more than half of patients, not only nonfamilial cases (showing a more benign prognosis with lower disease risk in family members), remain without a genetic answer [69,70,71]. 

Possible explanations of an inconclusive or negative genetic result could be multiple rare pathogenic variants inherited together (a polygenic disease model) or the combination of low effects of common variants with nongenetic factors (for example, age, hypertension, and obesity) (a complex disease model) [8,70,71]. Recently, a GWAS study of 2780 HCM cases and 47,486 controls identified 12 significant susceptibility loci for HCM. The authors also identified, in sarcomere-negative HCM cases, a strong polygenic influence and diastolic blood pressure as a major modifiable risk factor [72]. Another genome-wide study clearly demonstrated that the polygenic risk score can explain the phenotypic variability in HCM patients carrying causative rare variants [73]. Based on this new evidence, although HCM was mainly considered an autosomal dominant Mendelian disease, it is now gradually more recognized to have a complex genetic inheritance [68,74,75]. Moreover, genetic modifiers such as DNA methylation, acetylation, and miRNA have been demonstrated to have a role in HCM manifestation [76,77,78,79,80]. These new findings will be helpful to pave the way for a better understanding of the molecular pathophysiology of HCM and to try to cover the lack of current knowledge in the genotype—fibrotic phenotype correlation.

## 7. Conclusions and Future Research Scenarios

Myocardial fibrosis in HCM has traditionally been considered a secondary adaptation in response to hypertrophy; however, several studies have suggested that profibrotic signalling is detectable in the early stages of the disease before hypertrophy is evident [24,26]. While it is clear that cFB activation and differentiation into MyoFB with excessive collagen deposition are key events for the development of myocardial fibrosis in HCM, the link between sarcomere mutations and cardiac fibrosis has only been made by a few studies describing a cross-talk between affected cardiomyocytes and cFBs [19,20]. In this regard, in contrast to previous knowledge, recent evidence has also demonstrated that cFB does express sarcomeric proteins that are typically mutated in HCM, underlining the importance of considering both the cardiomyocytes and the cFB for the development of HCM [62]. Following this idea, single-cell RNA sequencing has allowed for a better insight into the interactions between the cardiomyocytes and cFB during the last few years and these studies have also been able to identify new key genes involved in HCM myocardial fibrosis [20]. Moreover, a complex genetic inheritance has emerged for HCM [68,74,75]. Undeniably, the routine application of genetic testing, with the preclinical identification of family members at risk of disease represents an important advance. However, to improve HCM patient management, personalized medicine will be more and more needed, and it must be focused not only on the identification of new genes and new variants but also on their interaction with the genetic background, epigenetic, and environmental factors. Taken together, these findings will pave the way to a better understanding of the cellular and genetic actors involved in the pathophysiology of HCM. 

Nonetheless, myocardial fibrosis in HCM remains a complex issue, and a more detailed knowledge of the molecular pathways involved at the different stages of HCM is demanded in order to develop specific therapeutic strategies. This is highly relevant considering that myocardial fibrosis was associated with severe arrhythmic outcomes in HCM patients in different studies, well summarized in a systematic review by Bittencourt M.I., et al. [81]. Of note, beyond the echocardiography, cardiac fibrosis is now evaluated also by cardiac magnetic resonance with late gadolinium enhancement, which highlights a regional increase in collagen content, and by T1 mapping, which detects a more diffuse pattern of fibrosis [82], helping the physicians to have a more comprehensive overview to recognize HCM patients at higher risk. 

Due to its central role in cardiac fibrosis, TGF-β and its downstream effectors are attractive pharmaceutical targets. Nevertheless, pharmacologic treatment of fibrosis has been challenging for several reasons, including the pleiotropic effects of TGF-β on different cell types [34]. Indeed, while sustained TGF-β signalling leads to excess ECM deposition with deleterious effects on cardiac function, animal studies have shown that complete abolition of TGF-β leads to ECM degradation with increased ventricular dilation, cardiac dysfunction, and animal mortality [50,52]. Conversely, the downregulation of TGF-β expression by Pirfenidone has been suggested to have antifibrotic activity and cardioprotective effects [36,83]. The efficacy and safety of Pirfenidone have been tested in heart failure patients enrolled in a double-blind phase 2 clinical trial with promising preliminary results as described by several authors [36,83,84]. Pharmaceutical treatment would therefore have to be finely balanced, specifically targeted, and delivered to the cardiac tissue [34,85]. Mechanotherapeutic strategies have also been attempted, suggesting that acting on molecular mechanobiology might result in a possible shutdown of fibrotic progression [86]. 

Finally, considering that fibrosis is part of the pathological disease process in HCM, finding a way to prevent myocardial fibrotic deposition would definitely be a promising avenue to ameliorate the clinical status of HCM patients.

## Figures and Tables

**Figure 1 ijms-24-14845-f001:**
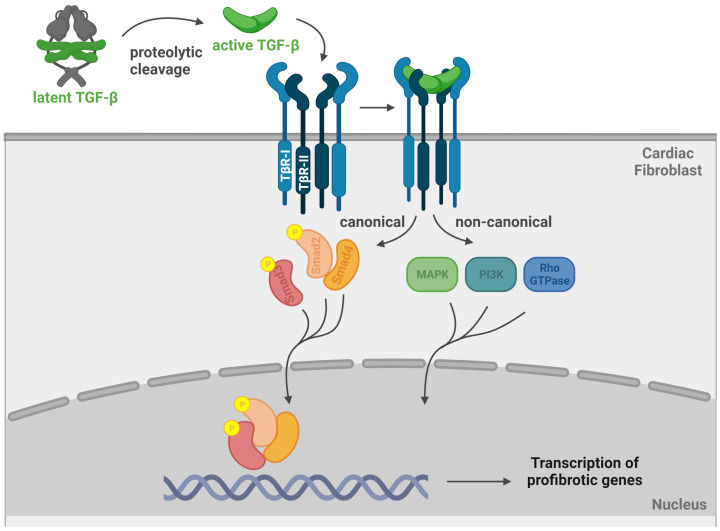
TGF-β signalling in cardiac fibroblasts. TGF-β is activated by proteolytic cleavage from the latent complex. Binding to the heterodimeric TGF-β receptor (TβR-I/TβR-II) activates profibrotic signalling cascades via canonical or non-canonical signalling pathways. The canonical pathway involves the phosphorylation of Smad2 and Smad3 and the association with Samd4. The complex translocates to the nucleus and regulates the transcription of profibrotic genes. Non-canonical TGF-β signalling comprises several different pathways, including mitogen-activated protein kinase (MAPK), phosphoinositide-3-kinase (PI3K), and Rho GTPases (Rho) pathways, which also alter the expression of fibrotic genes. Created with BioRender.com.

**Table 1 ijms-24-14845-t001:** Genes associated with HCM.

Gene	Protein	Function
** *ACTC1* **	**Actin Alpha Cardiac Muscle 1**	**Thin filament protein**
*ACTN2*	Actinin Alpha 2	Z lines
** *ALPK3* **	**Alpha Kinase 3**	**Sarcomere-associated protein**
*ANKRD1*	Cardiac Ankyrin Repeat Protein 1	Sarcomere-associated protein
*CALR3*	Calreticulin 3	Calcium homeostasis
*CAV3*	Caveolin 3	Sarcomere-associated protein
*CSRP3*	Cysteine And Glycine Rich Protein 3	Z lines
*FLNC*	Filamin C	Sarcomere-associated protein
*JPH2*	Junctophilin 2	Calcium homeostasis
*KLF10*	Kruppel-like Factor 10	Transcription factor
** *MYBPC3* **	**Myosin-binding protein C3**	**Thick filament protein**
*MYH6*	Myosin Heavy Chain 6	Thick filament protein
** *MYH7* **	**Myosin Heavy Chain 7**	**Thick filament protein**
** *MYL2* **	**Myosin Light Chain 2**	**Thick filament protein**
** *MYL3* **	**Myosin Light Chain 3**	**Thick filament protein**
*MYLK2*	Myosin Light Chain Kinase 2	Sarcomere-associated protein
*MYOM1*	Myomesin 1	M line
*MYOZ2*	Myozenin 2	Z lines
*MYPN*	Myopalladin	Sarcomere-associated protein
*NEXN*	Nexilin F-Actin Binding Protein	Sarcomere-associated protein
*OBSCN*	Obscurin, Cytoskeletal Calmodulin and Titin-Interacting RhoGEF	M line
*PDLIM3*	Alpha-Actinin-2-Associated LIM Protein	Sarcomere-associated protein
*PLN*	Phospholamban	Calcium homeostasis
*RYR2*	Cardiac Ryanodine Receptor 2	Calcium homeostasis
*TCAP*	Titin-Cap	Z lines
*TNNC1*	Troponin C1, Slow Skeletal And Cardiac Type	Thin filament protein
** *TNNI3* **	**Troponin I3, Cardiac Type**	**Thin filament protein**
** *TNNT2* **	**Troponin T2, Cardiac Type**	**Thin filament protein**
** *TPM1* **	**Tropomyosin 1**	**Thin filament protein**
*TRIM63*	Tripartite Motif Containing 63	M line
*TTN*	Titin	Thick filament protein
*VCL*	Vinculin	Sarcomere-associated protein

The nine most frequently mutated genes in HCM patients are highlighted in bold. These nine sarcomeric genes are indicated as definitive HCM genes by the ClinGen consortium (https://clinicalgenome.org/, (accessed on 1 June 2023) [13]. For more details on genes involved in syndromic HCM not reported here, see the following references [8,13].

**Table 2 ijms-24-14845-t002:** Profibrotic genes in HCM cardiac fibrosis.

Reference	Model	Genes [Method]
[19]	Non-myocyte cells isolated from ventricles of an HCM mouse model	50 cell-cycle genes; 44 ECM genes, 9 out of 44 are Tgfβ responsive genes (Postn, Eln, Dusp4, Tnc, Timp1, Col1a2, Col3a1, Col1a1, Ltbp2) [DSAGE] Ctgf, Tgfb1, and Tgfb2 [DSAGE and RT-PCR] Postn [DSAGE, RT-PCR, WB] pSmad2 [Immunohistochemistry]
[65]	Ventricles from HCM mouse model	miR-1, miR-30, miR-133, [TaqMan Low-Density Arrays, RT-PCR] Ctgf, Thbs1, serpinE1, Mtpn, Slc2A4, Fndc5 [Affymetrix GeneChip^®^ Mouse Gene 1.0 ST Arrays, RT-PCR]
[54]	Total heart and activated myofibroblasts in Mybpc3^40kDa^ transgenic mice	*Tgfb2, T gfb3, Bmp7* [RT-PCR, WB] *p-smad3, p-tak1, p38, p-smad1/5/9* [WB] *Ctgf* [RT-PCR] *Mmp14* [RT-PCR] *Timp1* [RT-PCR] *α-SMA* [WB] Periostin [WB, immunostaining] *TβRII* [RT-PCR, WB]
[66]	Serum samples from HCM patients with or without fibrosis	miR-29a, MIAT [RT-PCR]
[18]	Left ventricle from two mouse models of HCM and septal myectomy of HCM patients	*Tgfb1*, miR-29a/b/c, *Col1a1, Col1A2, Col3a1, eln* (in TnT mouse model) [RT-PCR] Collagen 1 *(in TnT mouse model)* [WB] *Tgfb2, Ctgf (in both TnT and MyHC mouse models)* [RT-PCR] *TGFB2, TGFB3, TGFBR2, ACE2, RENBP, IGF2R, AGT, AGTR1 (HCM patients)* [RNA-Seq]
[46]	HCM myectomy tissues, myocardial cells and interstitial fibroblasts	*FBLN2 (fibulin-2)* [immunostaining, WB]
[64]	Human cardiac-activated fibroblasts, from patients with advanced HCM	*COL22A1* [snRNA-Seq + RNA in situ + KO] *POSTN, SLC44A5, JAZF1, AEBP1, THBS4, ITGA10, ENPP1, FN, CLSTN2, PRELP, KCNMA1, PRRX1, CRISPLD2, FBLN5, COL4A4, NEGR1, C7* [snRNA-Seq + KO] *NOX4, FAP, COL1A1, COL1A2, FAM155A, TSHZ2, TMEM87B, ARHGAP42, COL4A3, FHIT, PDE1A, FBLN1, HIP1, CDH11, PDGFRA* [snRNA-Seq] *ACTA2, TGFBR1, TGFBR2, SMAD2, SMAD3* [KO]
[68]	myectomy tissue from HCM patients	*HIF-1α, IGFR-1, JAK1, JAK2* [RNA-Seq] *STAT3* [WB] *COL4A2* [RNA-Seq, WB]
[20]	Human HCM myectomy tissues (interventricular septum): cardiomyocytes^©^, fibroblasts (F) and lymphocytes (L)	*ACTC1, JUNB, MS4A6A, MYH7, NEAT1, TNNI3, COL6A1, COL6A2, COL1A2, TIMP1, COL3A1, FN1, LAMA2, CALM2, MFGE8, LPL, PSAP, SERPING1, VEGFA, SORBS1, ADAM17, HSPG2, LGALS3BP* [snRNA-Seq, in C] *C1R, C1S, C7, CILP, COL1A1, COL6A1, COL6A2, HES1, IGFBP7, MFAP4, MYL2, POSTN, SERPINF1, TNNT2, ITGB1, COL1A2, LRP1, C3, CALR, COL18A1, COL4A1, COL5A1, COL5A2, CXCL12, CYR61, FBLN1, FBN1, HSP90B1, IGF1, LAMA2, LAMB1, LAMC1, LGALS3BP, LRPAP1, NID1, PDGFD, PTN, TFPI, THBS2, TIMP2, TNC, COL3A1, COL6A3, FN1, HSPG2, VCAN* [snRNA-Seq, in F] *ITGB1* [snRNA-Seq, in L]
[21]	Left ventricles from HCM transgenic mice and HCM patients: lymphatic endothelial cells (LEC) and fibroblasts (F)	*Lpar1* [snRNA-Seq, in mouse and human LEC and F] *Ccl21a* [immunostaining in mouse LEC] *Reln* [RNAscope^©^ in mouse LEC] *Col1a1, Col4a3, Col4a1, Tgfb2, Mfap5, Fap* [snRNA-Seq in mouse F] *RELN and CCL21* [snRNA-Seq, in human]
[58]	*MYBPC3* mutant pig and *MYBPC3-KO* mouse NIH3T3 fibroblasts	*Tgfb1, Col1a1, and* Acta2 [RT-PCR, WB, in mutant pig hearts] *Col1a1,* Acta2, *Tgfb1, Gck, Pfk, and Ldha* [RT-PCR, WB, in MYBPC3-KO NIH3T3 fibroblasts] *HIF-1α, p-p65, CCL2, IL-1β, IL-6, and TNF-α* [WB, in MYBPC3-KO NIH3T3 fibroblasts]

DSAGE (deep sequence analysis of gene expression); RT-PCR (RealTime-PCR); WB (WesternBlot); RNA-Seq (Total RNA sequencing); snRNA-Seq (single nucleus RNA sequencing); KO (knockout), RNAscope^©^ (RNA in situ hybridization).

## Data Availability

Not applicable.

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
