# Peer review of "Myocardial Fibrosis in Hypertrophic Cardiomyopathy: A Perspective from Fibroblasts"

_ijms, 2023, doi:10.3390/ijms241914845_

Round 1

Reviewer 1 Report

Authors need to update the review with more information and needs to offer new insights. Should also incorporate figures and tables.

Myocardial Fibrosis in Hypertrophic Cardiomyopathy: A Perspective from Fibroblasts

Overview

The review highlights the role of fibroblasts and TGF in HCM. The review is well structured and recently published articles are well referred and cited (Moderately).

Major Comments

1.     Abstract should be re-written, about 90% is repeated information and does not provide insights into what review is about and what can be expected.

2.     Should incorporate figures and more tables.

3.     Should provide “Future research prospects” on how these understandings can help diagnose and prevent HCM in young people.

4.     Conclusion is too general, does not provide definite highlights.

5.     Use more recent articles, only 24 out of 69 are after 2020.

6.     Few systematic review articles are also available related to this topic. Authors can use them to revise the manuscript accordingly.

Minor Comments

1.     Line 8: Abbreviate HCM in Line 8.

2.     Line 13: Abbreviate cardiac fibroblast here, not in Line 90.

3.     Line 80: Mention full form for TGF-β when mentioned first.

4.     Restrict the use of abbreviations. Abbreviations should only be used if the intended words are repeated more than thrice. For example: Line 96 and 263, MMPs. Only repeated twice, no need use abbreviation, should use full form. Check for the same throughout the manuscript. Line 96: No need to abbreviate TIMPs.

Remark

The information provided well written and provides the basic information, however lack of figures and sufficient tables is a setback. The review for this topic requires more comprehensive literature presentation, which is lacking. Authors are requested to update the review with more recent findings (Articles relevant to this topic are also available in 2023).

Author Response

Here we detailed point-by-point answers to all comments.

Major Comments

  1. Abstract should be re-written, about 90% is repeated information and does not provide insights into what review is about and what can be expected.

As suggested, the abstract has been revised and modified. See lines 11-17.

  1. Should incorporate figures and more tables.

We have added the new Table 1 describing the known HCM genes, as also requested by Reviewer 1, and the Figure 1 describing TGF-β signaling in cardiac fibroblasts. See lines 49-53 and 258-269 respectively.

  1. Should provide “Future research prospects” on how these understandings can help diagnose and prevent HCM in young people.

In response to this comment and comment number 4 below, we have modified the Conclusion paragraph by inserting new text that emphasizes the possible future research scenario based on new knowledge with the aim of improving the management of patients with HCM. In addition, we have indicated the main highlights. Please see lines 361-391.

  1. Conclusion is too general, does not provide definite highlights.

Please, see answer to the above comment number 3.

  1. Use more recent articles, only 24 out of 69 are after 2020.

We have replaced and added newer articles to those previously cited. We now have about half of the post-2020 references, with many more from 2018 and 2019 (about 16). Some articles were not replaced because they described a very specific topic. See the articles highlighted in red in the References section.

  1. Few systematic review articles are also available related to this topic. Authors can use them to revise the manuscript accordingly.

As suggested, we have revised the manuscript with some systematic review articles and updated the references accordingly. Please see references (5, 67, 81):

  • Aziz, A.; Musiol, S.K.; Moody, W.E.; Pickup, L.; Cooper, R.; Lip, G.Y.H. Clinical prediction of genotypes in hypertrophic cardiomyopathy: A systematic review. Eur J Clin Invest. 2021, 51, e13593.
  • Scolari, F.L.; Faganello, L.S.; Garbin, H.I.; Piva E Mattos, B.; Biolo, A. A systematic review of microRNAs in patients with hypertrophic cardiomyopathy. Int J Cardiol. 2021, 327, 146-154.
  • Bittencourt, M.I.; Cader, S.A.; Araújo, D.V.; Salles, A.L.F.; Albuquerque, F.N.; Spineti, P.P.M.; Albuquerque, D.C.; Mouril-he-Rocha, R. Role of Myocardial Fibrosis in Hypertrophic Cardiomyopathy: A Systematic Review and Updated Meta-Analysis of Risk Markers for Sudden Death. Arq Bras Cardiol. 2019, 112, 281-289.

Reviewer 2 Report

In the review article ‘Myocardial Fibrosis in Hypertrophic Cardiomyopathy: A Perspective From Fibroblasts’ submitted by Maja Schlittler and coworkers to IJMS, the authors have presented a nice very well written article summarizing the major facts about cardiac fibrosis focussed on hypertrophic cardiomyopathy (HCM).

I suggest some minor changes:

1.)    It would be great if the authors can give a little bit more background on the genetic etiology of HCM. For example the authors can reference the following book chapter: Gerull, Brenda, Sabine Klaassen, and Andreas Brodehl. "The genetic landscape of cardiomyopathies." Genetic causes of cardiac disease (2019): 45-91.

2.)    Please write all gene names like MYH7 and MYBPC3 in Italics.

3.)    The TGFß signalling and the fibrotic remodelling is nicely discussed. The authors compared it also with scar formation after myocardial infarct. This is fine. However, it would be great if the authors can compare this also with the fibrotic remodelling in other cardiomyopathies. Recently, a DSC2 (desmocollin-2) transgenic mouse model was developed for arrhythmogenic cardiomyopathy (ACM). Of note, a severe fibrotic (and inflammatory) response was found in this mouse model. Could you outline shortly differences and similarities in the fibrotic remodelling of ACM and HCM?

4.)    Page 3, Line 136: Is the sheddase known for proteolytic TGFß cleavage?

5.)    Is there any clinical perspective for example by inhibiting the fibrotic pathways in HCM?

Author Response

Here we detailed point-by-point answers to all comments.

Minor Comments

  1. It would be great if the authors can give a little bit more background on the genetic etiology of HCM. For example the authors can reference the following book chapter: Gerull, Brenda, Sabine Klaassen, and Andreas Brodehl. "The genetic landscape of cardiomyopathies." Genetic causes of cardiac disease (2019): 45-91.

We have provided additional information on the genetic etiology of HCM in the Introduction section, based on, but not limited to, the book chapter suggested by the reviewer. New Table 1, describing the known HCM genes, has also been added. See lines 30-53, highlighted in red, in the Introduction section.

  1. Please write all gene names like MYH7 and MYBPC3 in Italics.

We have corrected all gene names in Italics.

  1. The TGFß signalling and the fibrotic remodelling is nicely discussed. The authors compared it also with scar formation after myocardial infarct. This is fine. However, it would be great if the authors can compare this also with the fibrotic remodelling in other cardiomyopathies. Recently, a DSC2 (desmocollin-2) transgenic mouse model was developed for arrhythmogenic cardiomyopathy (ACM). Of note, a severe fibrotic (and inflammatory) response was found in this mouse model. Could you outline shortly differences and similarities in the fibrotic remodelling of ACM and HCM?

As suggested by the Reviewer, we have briefly described differences and similarities in the fibrotic remodelling of ACM and HCM. Please see lines 74-81 in the revised manuscript.

  1. Page 3, Line 136: Is the sheddase known for proteolytic TGFß cleavage?

The A Disintegrin and Metalloproteinases ADAM10 and ADAM17 and matrix metalloproteinases are among the most known sheddases. They are enzymes that cleave extracellular portions of transmembrane proteins, releasing the soluble ectodomains from the cell surface. Receptor substrates of sheddases include among the others the cytokine receptors TNFR1, IL-6R and type-I and -III TGF-β receptors. In different context it has been described that the latent TGF-β complex is cleaved by a wide range of proteases, including thrombospondin 1, matrix metalloproteinases (MMP 2, MT1-MMP, ADAMTS) and activated integrins, such as αvβ6 and αvβ8 (See ref. 35,36 in the revised manuscript). Less is known on the proteolytic TGFß cleavage specifically in cardiac context. We have found some papers describing Furin, ADAMTS16, ADAMTS4, and LTBPs as activating factors of latent TGF-β in cardiac fibroblasts (See ref. 37-40 in the revised manuscript).

We have added in the revised manuscript the lines 154-161 on this topic.

  1. Is there any clinical perspective for example by inhibiting the fibrotic pathways in HCM?

To date, no clinical therapies are available to effectively block cardiac fibrosis and scientific gap of knowledge remain to be filled on the HCM specific myocardial fibrosis, before to move to possible novel specific therapeutic actions. For example try to inhibit the fibrotic pathways by modulating the TGF-β signalling may cause several adverse effects, as TGF-β has pleiotropic properties on different cell types. Indeed, many strategies blocking TGFβ signaling pathways have been explored in animal models, leading to an increased cardiac dilatation and animal mortality (See ref. 50,52 in the revised manuscript). For this reason  pharmacologic treatment of fibrosis by acting on TGF-β could be extremely challenging. However, several evidence indicate that Pirfenidone has an anti-fibrotic activity and cardioprotective effects (See ref. 36,83 in the revised manuscript), based also on promising data in heart failure patients treated with this drug in a double-blind phase 2 clinical trial (See ref. 36,83,84 in the revised manuscript). Considering that the fibrosis is part of the pathological disease process in HCM, to find a way to prevent fibrotic deposition would be definitely a promising perspective to ameliorate the clinical status of HCM patients. We have accordingly modified the revised manuscript. See lines 366-391.
